# Real-Life Characteristics of Patients with COPD and Discordant CAT-mMRC Questionnaires

**DOI:** 10.3390/jcm14248771

**Published:** 2025-12-11

**Authors:** Andrea Portacci, Vitaliano Nicola Quaranta, Alessio Marinelli, Alessandro Capuano, Maria Rosaria Vulpi, Fabrizio Diaferia, Carla Santomasi, Mariafrancesca Grimaldi, Giovanni Sanasi, Marianna Cicchetti, Eustachio Ricciardi, Giulia Amoruso, Alfredo Vozza, Silvano Dragonieri, Giovanna Elisiana Carpagnano

**Affiliations:** 1Department of Translational Biomedicine and Applied Neuroscience, Institute of Respiratory Disease, University of Bari “Aldo Moro”, 70121 Bari, Italy; a.portacci01@gmail.com (A.P.); vitalianonicola.40@gmail.com (V.N.Q.); alessio.marinelli@uniba.it (A.M.); a.capuano13@studenti.uniba.it (A.C.); mariarosaria86@alice.it (M.R.V.); fabriziodiaferia@gmail.com (F.D.); carlasantomasi@gmail.com (C.S.); mariafrancescagrimaldi@gmail.com (M.G.); giovannisanasi@libero.it (G.S.); mariannacic@gmail.com (M.C.); eus.ricciardi@gmail.com (E.R.); giulia.amoroso@uniba.it (G.A.); elisiana.carpagnano@uniba.it (G.E.C.); 2Department of Precision and Regenerative Medicine and Ionian Area, University of Bari “Aldo Moro” 70121 Bari, Italy; alfredo.vozza@uniba.it

**Keywords:** COPD, CAT, mMRC, patient-reported outcomes, exacerbations

## Abstract

**Background/Objective**: The Global Initiative for Chronic Obstructive Lung Disease (GOLD) recommends using the COPD Assessment Test (CAT) and the modified Medical Research Council (mMRC) scale for symptom assessment. However, discordant results between these tools are common and may affect clinical phenotyping and treatment decisions. This study aims to identify clinical, functional, and radiological characteristics associated with discordant CAT and mMRC scores in COPD and assess the stability of this classification over time. **Methods**: We retrospectively analyzed 222 COPD patients classified into GOLD A, B, E, or a newly defined discordant group (GOLD_D_), characterized by CAT ≥ 10 with mMRC < 2 or CAT < 10 with mMRC ≥ 2. Clinical, functional, laboratory, and imaging data were collected. Logistic regression identified predictors of GOLD classification. GOLD group changes were assessed at follow-up. **Results**: At baseline, 12.8% of patients belonged to the GOLD_D_ group. Compared to GOLD A and B, these patients had lower occupational exposure and higher rates of chest HRCT findings such as consolidations and centrilobular nodules. Regression models confirmed these features and identified FEV1 and FVC as independent predictors. GOLD classification showed notable variability during follow-up. **Conclusions**: Patients with discordant CAT-mMRC scores display distinct clinical and radiologic traits not captured by standard GOLD categories. These results underscore the limitations of relying solely on symptom scores and support a more comprehensive, trait-based approach to COPD assessment and management.

## 1. Introduction

Chronic obstructive pulmonary disease (COPD) is a prevalent and debilitating respiratory condition characterized by persistent limitations in airflow and a range of respiratory symptoms [1]. COPD has established itself as a preeminent cause of global mortality, currently ranking as the third leading cause of death worldwide. The epidemiological burden of this disease indicates that the pathology claimed over 3 million lives, representing 6% of aggregate global mortality [1].

Despite being clinically defined as both preventable and treatable, COPD remains a critical public health challenge characterized by significant chronic morbidity. The disease trajectory often involves prolonged physiological decline, resulting in premature mortality either directly from the condition or its associated sequelae. Furthermore, predictive models suggest an escalating global burden in the forthcoming decades; this trajectory is attributed to demographic shifts toward an aging population and persistent exposure to established risk factors.

COPD presents significant clinical and functional heterogeneity, demanding a comprehensive evaluation for optimal diagnosis, treatment strategies and effective patient stratification. Recognizing this complexity, the Global Initiative for Chronic Obstructive Lung Disease (GOLD) strategy emphasizes the importance of integrating lung function assessments, symptom evaluation, and exacerbation history to facilitate personalized patient care in COPD [1]. Among the recommended tools for evaluating symptom burden, the COPD Assessment Test (CAT) and the modified Medical Research Council (mMRC) dyspnea scale are essential for assessing disease impact and guiding treatment decisions [2]. The CAT is a multifaceted questionnaire encompassing eight items addressing various symptoms, including cough, shortness of breath, sputum production, chest tightness, capacity for daily activities, self-confidence, sleep quality, and energy levels [3]. Conversely, the mMRC scale evaluates specifically the degree of dyspnea experienced by the patient using a five-point grading system [4]. This fundamental difference in scope and structure between the two instruments could lead to discrepancies in the resulting scores for individual patients. The 2024 GOLD report reinforces the role of both CAT and mMRC in the initial evaluation of COPD. However, in routine clinical practice, discordant results are frequently observed when both questionnaires are administered to the same patient [5]. This variability in patient responses raises critical questions regarding the optimal application and interpretation of these tools. Furthermore, the temporal stability of this classification has often been overlooked, raising concerns about its reliability during follow-up assessments. The aim of this study is to evaluate what clinical, functional, laboratory and radiological features characterize patients showing discordant CAT and mMRC scores according to GOLD classification.

## 2. Methods

### 2.1. Study Design

We conducted a real-life, observational, retrospective study enrolling patients with a confirmed diagnosis of COPD according to GOLD recommendations from February 2020 to December 2023.

### 2.2. Study Population

All the patients who visited our COPD clinic underwent a complete clinical evaluation, including anthropometric and anamnestic information, comorbidities assessment, respiratory symptoms, GOLD stage, number and severity of COPD exacerbations. Lung function assessments were performed using flow-volume spirometry and/or body plethysmography (Masterlab Jaeger, Höchberg, Germany) following ERS/ATS standards [6]. We also routinary performed an arterial blood gas analysis (ABG) and a 6 min walking test (6MWT) to evaluate gas exchanges and exercise tolerance. Finally, according to the clinical judgment, clinicians could require a chest high resolution computed tomography (HRCT). After a complete screening of the electronic medical records, 222 patients were selected for the final analysis. Patients were excluded when CAT or mMRC score were not available. The study was approved by the Bari Institutional Ethics Committees (Ethical Committee number: 1709/CEL—12 June 2024) and was conducted following the Helsinki Declaration of 1975 and the Good Clinical Practice standards. Patients signed written informed consent before enrollment.

### 2.3. GOLD Classification

At the baseline visit, patients were classified in the GOLD A group when scoring a CAT < 10 and a mMRC < 1, while patients reporting a CAT ≥ 10 and a mMRC ≥ 2 were grouped as GOLD B. Patients with at least two moderate exacerbations or 1 exacerbation leading to hospitalization in the year before the enrollment were considered in the GOLD E group. Then, we created a new group called GOLD_Discordant_ (GOLD_D_) for those patients with CAT ≥ 10 and a mMRC < 2 or with CAT < 10 and mMRC ≥ 2 (Table 1).

### 2.4. Statistical Analysis

Continuous variables distribution was assessed using Shapiro–Wilk test. Then, we described variables with Gaussian distribution as means ± standard deviations (SD), while those with non-normal distribution were evaluated as medians and interquartile ranges (IQR). Student *t*-test, Mann–Whitney test and Wilcoxon test were then used for comparisons. We expressed binary variables as percentages, using Fisher exact test for comparisons. Spearman coefficients were calculated for correlation analysis. To identify variables that differentiate between GOLD_D_ and GOLD A-B groups, we conducted a multiple backward logistic regression analysis. For each model, we evaluated robustness using receiver operating characteristic (ROC) curves, along with the corresponding area under the curve (AUC) and odds ratios (OR). We developed three models comparing GOLD_D_ with GOLD A (model 1), GOLD B (model 2), and the combined GOLD A-B cohort (model 3). In all these models, the classification as GOLD_D_ was considered as dependent variable. Independent covariates that showed statistically significant differences in the preliminary descriptive analysis were included in the models. We used SPSS Statistics 26 (IBM Corporation, Armonk, NY, USA ) and R software (version 4.0.2, R Foundation, Vienna, Austria) for statistical purposes, considering a *p* value < 0.05 as statistically significant.

## 3. Results

### Baseline Features

From February 2020 to December 2023, 222 patients with a diagnosis of COPD were enrolled and completed both the CAT and mMRC questionnaires. Among the enrolled cohort, 55 patients (24.8%) were classified as belonging to GOLD A group, 76 (34.2%) to group B, 63 (28.4%) to group E, and 28 (12.6%) to the discordant questionnaires group (GOLD_D_, as shown in Figure 1). There were no deaths recorded in any of the groups throughout the study.

Comparisons of baseline clinical and functional features in patients classified as GOLD A, GOLD B, and GOLD_D_ are summarized in Table 2 and Appendix A. Patients belonging to GOLD A and GOLD B classes reported more frequent exposure to work toxics and pollutants compared to GOLD_D_ patients (*p* = 0.002 and <0.0001, respectively). Moreover, GOLD B group showed a significant higher body mass index (BMI, *p* = 0.001) and a worse lung function than GOLD_D_ patients.

In the enrolled cohort, 159 patients (71.6%) underwent chest HRCT during their baseline visit. Patients in the GOLD E group were significantly more likely to undergo chest HRCT (*p* = 0.01). Among the subgroups, patients classified as GOLD_D_ showed a higher prevalence of lung consolidations and multiple centrilobular nodules compared to all other groups (*p* = 0.01 and *p* = 0.03, respectively; see Table 3 and Appendix A). As part of the baseline visit, all the enrolled patients performed ABG analysis and a 6MWT. We found no significant differences in ABGs parameters comparing GOLD A, GOLD B, and GOLD_D_ patients, while patients with GOLD E had worse mean PaO2 values in the overall cohort (see Table 4 and Appendix A, *p* < 0.0001). 6MWT revealed worse pre-test SpO2 (*p* = 0.006) and 6MWD (*p* = 0.002) in patients belonging to GOLD B group compared to those included in GOLD D.

To identify which clinical, functional, and radiological features could better define patients included in GOLD_D_ group, we developed three multivariate backword logistic regression models (Table 5). Models 1 and 2 directly compared GOLD D with GOLD A and GOLD B groups, respectively, while the third model explored included the combined subgroup of GOLD A-B patients. In model 1, the exposure to work pollutants (OR 0.03, *p* = 0.009) and the presence of consolidations (OR 7.8, *p* = 0.04) or centrilobular nodules (OR 4.4, *p* = 0.03) from chest HRCT were significantly related to GOLD D group. In Model 2, the exposure to work pollutants (OR 0.02, *p* = 0.002), the presence of chest HRCT consolidations (OR 13.7, *p* = 0.02) and an increased FEV1 (OR 8.7, *p* = 0.001) better differentiated GOLD D from GOLD B group. In Model 3, we confirmed the significance of exposure to occupational pollutants (OR = 0.03, *p* = 0.005), the presence of HRCT-detected consolidations (OR = 5.9, *p* = 0.03), and centrilobular nodules (OR = 3.5, *p* = 0.03) as independent predictors of GOLD D group membership. Additionally, FVC was also identified as a statistically significant covariate (OR = 2.3, *p* = 0.01). All the three models resulted robust after ROC analysis (see Figure 2, *p* < 0.0001).

Finally, we assessed the stability of the GOLD classification at follow-up using a Sankey plot (Figure 3). At baseline, most of the patients were classified as GOLD B (36.8%) and GOLD E (32.6%), while smaller proportions fell into GOLD A (15.8%) and GOLD D (14.7%). Among those initially in GOLD_D_, 35.7% continued to exhibit discordant CAT and mMRC scores and another 35.7% were reclassified into GOLD A. By contrast, fewer patients worsened their clinical condition, moving into GOLD B (21.5%) or GOLD E (7.1%). At the follow-up assessment, 16.8% of the cohort met the criteria for GOLD_D_; this group comprised five patients initially classified as GOLD A or B who subsequently developed CAT and mMRC scores discordance and one patient who transitioned from GOLD E to GOLD_D_.

## 4. Discussion

COPD is a complex disease characterized by diverse functional, radiological, and inflammatory alterations that significantly impact patients’ prognosis and quality of life. Collectively, these features define multiple disease phenotypes, which could be addressed through personalized therapeutic strategies [7]. Despite growing efforts to better understand COPD pathophysiology and disease evolution, the real-life applicability of the “treatable traits” approach remains uncertain [8]. Currently, COPD-related symptoms represent a pivotal trait in treatment algorithms. Apart from patients with frequent exacerbations (classified in GOLD group E), the use of patient-reported outcomes (PROs) such as the CAT and the mMRC dyspnea scale has become crucial [1]. However, these tools reflect only a fraction of the full complexity of COPD. The mMRC provides a unidimensional assessment of dyspnea, whereas the CAT includes multiple items related to daily symptoms and health-related quality of life. These differences help explain the often poor agreement between the two PROs, with some patients exhibiting a high CAT score but an mMRC < 2, and vice versa. In our study, we identified that 12% of the cohort fell into this discordant group, which we designated as GOLD_D_. Patients in this cluster showed distinct characteristics compared to those in GOLD A and B groups. As shown in Table 2, GOLD_D_ patients reported less frequent occupational exposure to toxic agents. Functionally, they exhibited greater impairment in FEV_1_ and FVC compared to GOLD A patients, while their decline was less severe than in GOLD B. HRCT findings also revealed a higher frequency of lung consolidations and multiple centrilobular nodules among GOLD_D_ patients, suggesting an increased susceptibility to infectious exacerbations relative to GOLD A and B.

These results were consistent with the multivariate regression analysis, which identified occupational exposure, specific HRCT abnormalities, and lung function impairment as the main factors distinguishing GOLDD patients from those in GOLD A-B. This pattern suggests that the discordance between CAT and mMRC may arise from different underlying mechanisms. Patients with high CAT scores but low mMRC values may be experiencing symptoms that are not primarily driven by ventilatory limitation such as cough, chest tightness, fluctuations in hyperinflation or sleep-related disturbances. In contrast, individuals with marked dyspnea despite a low CAT score might reflect a phenotype more influenced by exertional ventilatory constraints, dynamic hyperinflation or reduced inspiratory muscle efficiency. These possibilities deserve further study, as they may point to distinct pathophysiological profiles within the broader COPD population.

The poor agreement between CAT and mMRC score has been previously described in the literature. Kim and colleagues reported only a moderate agreement between these two measures (k = 0.51), with cough, sputum, chest tightness and sleep quality having the lowest level of correlation with mMRC dyspnea grading system [5]. In this study, only 2.7% of patients with CAT < 10 showed an mMRC score ≥ 2, while 33.1% of those with CAT ≥ 10 had an mMRC < 2. For this reason, the authors supported the idea of lowering mMRC cut-off to 1, as it could better identify patients with CAT ≥ 10 points. Mittal and Chhabra found slightly better results, with a substantial agreement between CAT and mMRC (k = 0.61) but with almost one-third of patients with mMRC ≥ 2 showing CAT < 10 [9]. Interestingly, using a greater CAT cutpoint (17 points) resulted in only a fair agreement between these two PROs (k = 0.4). Finally, Jones et al. [10] and Holt et al. [11] reported similar Cohen’s kappa coefficients in two independent cohorts (k = 0.62 and k = 0.63, respectively), concluding that the CAT identifies more symptomatic patients than the mMRC scale, likely due to its more comprehensive structure.

Regarding the clinical, radiological, and functional characterization of patients with discordant CAT-mMRC scores, Huang and colleagues reported wheezing as more frequent in patients with CAT ≥ 10 had an mMRC < 2 [12]. However, while the authors find no differences in lung function or comorbidities, they did not explore other features such as 6MWT, ABG analysis and chest HRCT. As also reported by GOLD recommendations, a thorough assessment of lung gas exchange, exercise capacity and chest imaging is of pivotal importance to detect COPD treatable traits [13]. Nevertheless, most therapeutic decisions are still based on the CAT-mMRC classification, which may fail to clearly identify specific phenotypes that could benefit from targeted treatment. In our study, we highlighted that patients classified within the GOLD_D_ group may exhibit distinct characteristics that are not captured by the traditional GOLD A-B categories. From a clinical perspective, relying only on PRO-based GOLD classification can be misleading. Patients in the GOLD_D_ group may be labelled as “low-risk” even when they show functional impairment or HRCT features that point to a higher likelihood of exacerbations. A broader assessment that includes imaging findings, lung function progression and exposure history can support more appropriate treatment decisions in this subgroup.

Another noteworthy aspect concerns potential changes in GOLD classification over the course of follow-up. In our cohort, several patients shifted GOLD categories between baseline and the first follow-up visit (see Figure 3). Specifically, 35.7% and 28.6% of patients initially classified as GOLD_D_ transitioned to GOLD A and B, respectively. Furthermore, 28.6% remained in the GOLD_D_ category, while only one patient was reclassified into group E. GOLD category shift during follow up visits could be related to many factors such as exacerbations development, treatment effect or risk factors exposure. Moreover, mMRC scale poorly discriminates changes in dyspnea severity over time [14]. While GOLD reports recommend considering ABE group system only for the initial therapeutic choice, it is interesting to note that several patients exhibited discordant CAT-mMRC scores even during the follow up.

The considerable shift in GOLD categories seen during follow-up shows how variable PRO-based classification can be over time. Symptom changes may depend on treatment effects, recent exacerbations, comorbid conditions, or even day-to-day variability in how patients report their symptoms. This dynamic pattern highlights the limits of fixed PRO thresholds and supports the use of broader, longitudinal assessments to better reflect how the disease evolves.

Our study has several limitations. The low sample size could have underestimated our results. This is particularly true for GOLD class changes analysis, since only 95 patients had at least one follow-up visit. Another limitation is the lack of specific radiological score to better understand the impact of chest HRCT alterations on clinical and functional variables. Finally, we did not evaluate the relationship between CAT-mMRC discordance and airway inflammatory biomarkers, which could have provided valuable insights into a potential endotypic fingerprint of GOLD_D_ patients.

## 5. Conclusions

In conclusion, our study offers a real-life perspective on the clinical, functional, and radiological characteristics of patients presenting with discordant CAT-mMRC scores. Future research should aim to elucidate the underlying pathophysiological mechanisms driving this discordance and explore tailored therapeutic strategies for this specific patient subgroup.

## Figures and Tables

**Figure 1 jcm-14-08771-f001:**
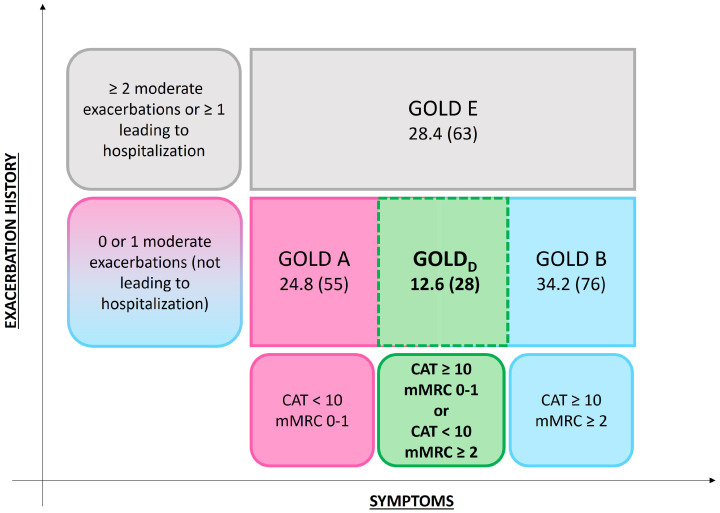
Distribution of COPD patients according to GOLD 2024 classification and discordant CAT and mMRC scores.

**Figure 2 jcm-14-08771-f002:**
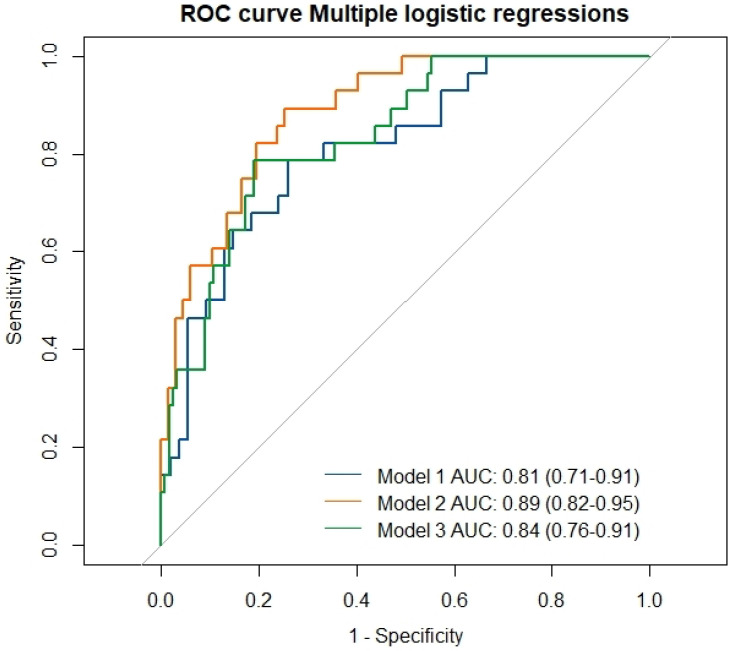
ROC curves for three logistic regression models identifying predictors of GOLD_D_ group classification.

**Figure 3 jcm-14-08771-f003:**
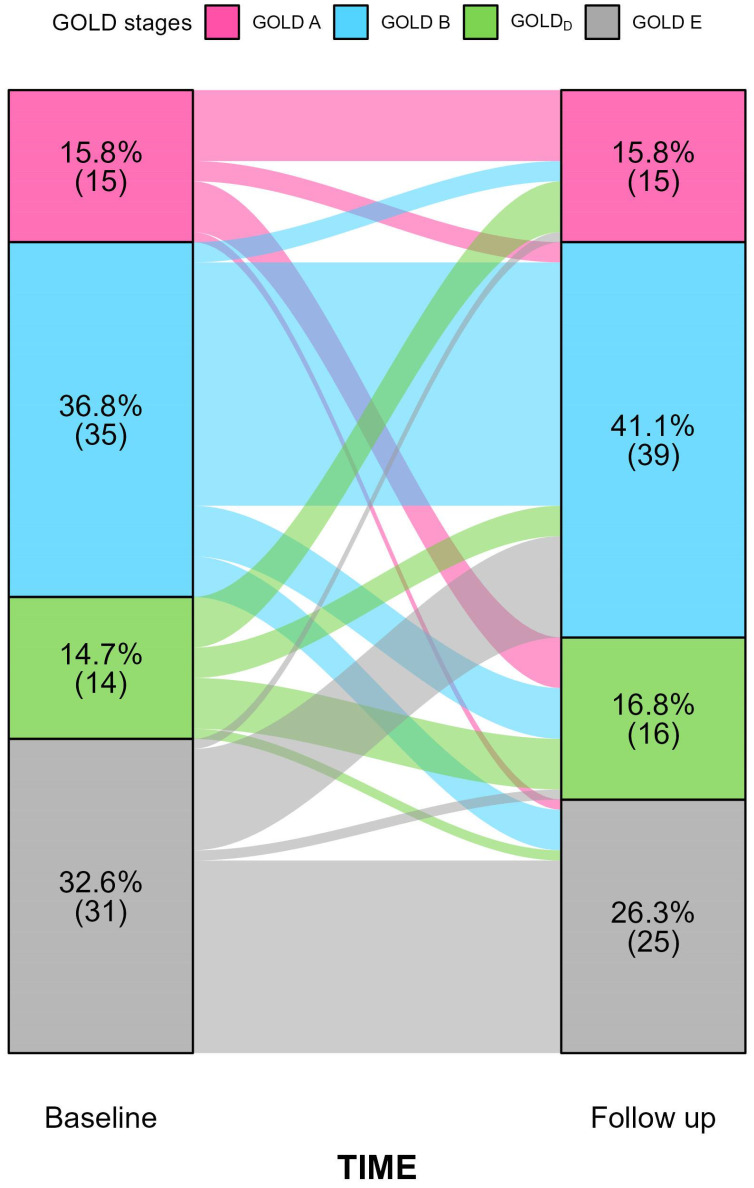
Sankey diagram illustrating transitions among GOLD A–E categories from baseline to follow-up in patients with complete paired data (N = 95). For each stratum, the upper value denotes the percentage of patients in that category and the value below the corresponding count. Coloured flows depict individual-level movement across GOLD stages over time.

**Table 1 jcm-14-08771-t001:** Modified GOLD classification.

Group	Criteria
GOLD A	CAT < 10 and mMRC < 1
GOLD B	CAT ≥ 10 and mMRC ≥ 2
GOLD E	≥2 moderate exacerbations or ≥1 severe exacerbation
GOLD_D_	CAT ≥ 10 and mMRC < 2
	CAT < 10 and mMRC ≥ 2

**Table 2 jcm-14-08771-t002:** Comparisons of clinical and functional features in patients with COPD: GOLD_D_ vs. GOLD A and GOLD_D_ vs. GOLD B stage.

	GOLD_D_	GOLD A	vs. *p*-Value	GOLD B	vs. *p*-Value
**Patients (n, %)**	12.6 (28)	24.8 (55)		34.2 (76)	
Age (Years, Median, IQR)	69.5 [63.7–77.2]	68 [61–74]	0.12	72 [63.2–79]	0.72
Gender (Male/Female, %)	92.9/7.1	78.2/21.8	0.12	80.3/19.7	0.15
BMI (kg/m^2^ Median, IQR)	25 [22.5–27.3]	26 [23–29]	0.25	29 [25–33.7]	**0.001**
**Smoke habits (n, %)**			0.35		0.32
Current smoker	28.6 (8)	43.6 (24)		44.7 (34)	
Former smoker	64.3 (18)	47.3 (26)		48.7 (37)	
No smoker	7.1 (2)	9.1 (5)		6.6 (5)	
**Atopy (n, %)**	21.4 (6)	20 (11)	0.99	18.4 (14)	0.78
Familiarity (n, %)	10.7 (3)	23.6 (13)	0.24	11.8 (9)	0.99
Work exposure (n, %)	3.6 (1)	32.7 (18)	**0.002**	38.2 (29)	**<0.0001**
**Comorbidities (n, %)**					
Arterial hypertension	50 (14)	70.9 (39)	0.09	64.5 (49)	0.26
Ischemic cardiomyopathy	25 (7)	12.7 (7)	0.22	31.6 (24)	0.63
Atrial fibrillation	17.9 (5)	9.1 (5)	0.29	15.8 (12)	0.77
Pulmonary hypertension	3.6 (1)	1.8 (1)	0.99	2.6 (2)	0.99
Asthma	0	0	/	5.3 (4)	0.57
GERD	3.6 (1)	12.7 (7)	0.26	10.5 (8)	0.44
OSAS	7.1 (2)	9.1 (5)	0.99	21.1 (16)	0.14
Cancer	21.6 (6)	23.6 (13)	0.99	15.8 (12)	0.56
CKD	3.6 (1)	5.5 (3)	0.99	11.8 (9)	0.28
Type2 Diabetes	28.6 (8)	20 (11)	0.42	19.7 (15)	0.42
Dyslipidemia	25 (7)	41.8 (23)	0.15	32.9 (25)	0.48
Thyroid disease	14.3 (4)	18.2 (10)	0.76	18.4 (14)	0.77
Depression	3.6 (1)	7.3 (4)	0.66	7.9 (6)	0.67
Total comorbidities (Median, IQR)	2 [1–3.7]	3 [1–4]	0.47	3 [1.2–5]	0.17
**Symptoms at first visit (n, %)**					
Dyspnea	82.1 (23)	81.8 (45)	0.99	91.6 (73)	**0.03**
Cough	50 (14)	52.7 (29)	0.82	52.6 (40)	0.83
Phlegm	46.4 (13)	50.9 (28)	0.82	51.3 (39)	0.82
Wheezing	0	1.8 (1)	0.99	0	/
Weight loss	0	0	/	3.9 (3)	0.56
**CAT (Median, IQR)**	11 [8.2–15.7]	6 [4–8]	**<0.0001**	16 [13–23]	**<0.0001**
**mMRC (Median, IQR)**	1 [1–2]	1 [1–1]	**<0.0001**	3 [2–3]	**<0.0001**
**Exacerbations (n, %)**	17.9 (5)	16.4 (9)	0.99	26.3 (20)	0.44
**Lung function**					
FEV1 (%, Mean, SD)	67.9 ± 18	75.5 ± 17.9	0.08	61.3 ± 18.1	**0.01**
FVC (%, Mean, SD)	88.2 ± 18	94 ± 17.9	0.19	77.9 ± 20.9	**0.01**
FEV1/FVC (%, Mean, SD)	55.3 ± 12	60.2 ± 6.9	0.06	58.4 ± 12.7	0.21
RV/TLC (%, Median, IQR)	127 [110–145.5]	122 [112–133]	0.29	135 [116–156.8]	0.07
**Treatments (%, n)**					
LAMA	14.3 (4)	34.5 (19)	0.007	7.9 (6)	0.45
LAMA + LABA	31.1 (9)	31.7 (18)	0.99	28.9 (22)	0.81
LAMA + LABA + ICS	42.9 (12)	18.2 (10)	**0.02**	44.7 (34)	0.99
OCS	3.6 (1)	1.8 (1)	0.99	2.6 (2)	0.99
LTOT	10.7 (3)	1.8 (1)	0.11	34.2 (6)	**0.02**
N-Acetylcysteine	3.6 (1)	5.5 (3)	0.99	9.2 (7)	0.68
	**GOLD_D_**	**GOLD A**	**vs.** * **p** * **-Value**	**GOLD B**	**vs. ** * **p** * **-Value**
**Patients (n, %)**	12.6 (28)	24.8 (55)		34.2 (76)	
Age (Years, Median, IQR)	69.5 [63.7–77.2]	68 [61–74]	0.12	72 [63.2–79]	0.72
Gender (Male/Female, %)	92.9/7.1	78.2/21.8	0.12	80.3/19.7	0.15
BMI (kg/m^2^ Median, IQR)	25 [22.5–27.3]	26 [23–29]	0.25	29 [25–33.7]	**0.001**
**Smoke habits (n, %)**			0.35		0.32
Current smoker	28.6 (8)	43.6 (24)		44.7 (34)	
Former smoker	64.3 (18)	47.3 (26)		48.7 (37)	
No smoker	7.1 (2)	9.1 (5)		6.6 (5)	
**Atopy (n, %)**	21.4 (6)	20 (11)	0.99	18.4 (14)	0.78
Familiarity (n, %)	10.7 (3)	23.6 (13)	0.24	11.8 (9)	0.99
Work exposure (n, %)	3.6 (1)	32.7 (18)	**0.002**	38.2 (29)	**<0.0001**
**Comorbidities (n, %)**					
Arterial hypertension	50 (14)	70.9 (39)	0.09	64.5 (49)	0.26
Ischemic cardiomyopathy	25 (7)	12.7 (7)	0.22	31.6 (24)	0.63
Atrial fibrillation	17.9 (5)	9.1 (5)	0.29	15.8 (12)	0.77
Pulmonary hypertension	3.6 (1)	1.8 (1)	0.99	2.6 (2)	0.99
Asthma	0	0	/	5.3 (4)	0.57
GERD	3.6 (1)	12.7 (7)	0.26	10.5 (8)	0.44
OSAS	7.1 (2)	9.1 (5)	0.99	21.1 (16)	0.14
Cancer	21.6 (6)	23.6 (13)	0.99	15.8 (12)	0.56
CKD	3.6 (1)	5.5 (3)	0.99	11.8 (9)	0.28
Type2 Diabetes	28.6 (8)	20 (11)	0.42	19.7 (15)	0.42
Dyslipidemia	25 (7)	41.8 (23)	0.15	32.9 (25)	0.48
Thyroid disease	14.3 (4)	18.2 (10)	0.76	18.4 (14)	0.77
Depression	3.6 (1)	7.3 (4)	0.66	7.9 (6)	0.67
Total comorbidities (Median, IQR)	2 [1–3.7]	3 [1–4]	0.47	3 [1.2–5]	0.17
**Symptoms at first visit (n, %)**					
Dyspnea	82.1 (23)	81.8 (45)	0.99	91.6 (73)	**0.03**
Cough	50 (14)	52.7 (29)	0.82	52.6 (40)	0.83
Phlegm	46.4 (13)	50.9 (28)	0.82	51.3 (39)	0.82
Wheezing	0	1.8 (1)	0.99	0	/
Weight loss	0	0	/	3.9 (3)	0.56
**CAT (Median, IQR)**	11 [8.2–15.7]	6 [4–8]	**<0.0001**	16 [13–23]	**<0.0001**
**mMRC (Median, IQR)**	1 [1–2]	1 [1–1]	**<0.0001**	3 [2–3]	**<0.0001**
**Exacerbations (n, %)**	17.9 (5)	16.4 (9)	0.99	26.3 (20)	0.44
**Lung function**					
FEV1 (%, Mean, SD)	67.9 ± 18	75.5 ± 17.9	0.08	61.3 ± 18.1	**0.01**
FVC (%, Mean, SD)	88.2 ± 18	94 ± 17.9	0.19	77.9 ± 20.9	**0.01**
FEV1/FVC (%, Mean, SD)	55.3 ± 12	60.2 ± 6.9	0.06	58.4 ± 12.7	0.21
RV/TLC (%, Median, IQR)	127 [110–145.5]	122 [112–133]	0.29	135 [116–156.8]	0.07
**Treatments (%, n)**					
LAMA	14.3 (4)	34.5 (19)	0.007	7.9 (6)	0.45
LAMA + LABA	31.1 (9)	31.7 (18)	0.99	28.9 (22)	0.81
LAMA + LABA + ICS	42.9 (12)	18.2 (10)	**0.02**	44.7 (34)	0.99
OCS	3.6 (1)	1.8 (1)	0.99	2.6 (2)	0.99
LTOT	10.7 (3)	1.8 (1)	0.11	34.2 (6)	**0.02**
N-Acetylcysteine	3.6 (1)	5.5 (3)	0.99	9.2 (7)	0.68

**Table 3 jcm-14-08771-t003:** Comparisons of radiological features in patients with COPD: GOLD_D_ vs. GOLD A and GOLD_D_ vs. GOLD B stage.

	GOLD_D_	GOLD A	vs. *p*-Value	GOLD B	vs. *p*-Value
**HRCT assessed (%, n)**	78.6 (22)	63.6 (35)	0.21	64 (48)	0.23
**HRCT findings (%, n)**					
Emphysema	39.3 (11)	36.4 (20)	0.81	40 (30)	0.99
GGO	10.7 (3)	3.6 (2)	0.33	8 (6)	0.7
Consolidations	21.4 (6)	3.6 (2)	**0.02**	4 (3)	**0.01**
Septal thickening	10.7 (3)	14.5 (8)	0.74	13.3 (10)	0.99
Lung scars	7.1 (2)	21.8 (12)	0.12	18.7 (14)	0.22
Bronchiectasis	14.3 (4)	12.7 (7)	0.99	13.3 (10)	0.99
Mosaic attenuation	3.6 (1)	1.8 (1)	0.99	4 (3)	0.99
Multiple nodules	32.1 (9)	10.9 (9)	**0.03**	10.7 (8)	**0.01**
Solitary nodules	14.3 (4)	14.5 (8)	0.99	16 (12)	0.99
Pleural effusion	0	0	/	1.3 (1)	0.99

**Table 4 jcm-14-08771-t004:** Comparisons of arterial blood gas analysis and 6 min walking test features in patients with COPD: GOLD_D_ vs. GOLD A and GOLD_D_ vs. GOLD B stage.

	GOLD_D_	GOLD A	vs. *p*-Value	GOLD B	vs. *p*-Value
**ABG**					
pH (median, IQR)	7.43 [7.41–7.44]	7.43 [7.42–7.44]	0.43	7.43 [7.41–7.45]	0.83
PaO_2_ (mmHg, mean, SD)	80.4±11.8	86±9.3	0.06	78.3±11.9	0.45
PaCO_2_ (mmHg, median, IQR)	39 [37.5–43]	39 [36–41.6]	0.39	40 [37–43.2]	0.93
HCO_3_^−^ (mEq/L, median, IQR)	26 [25–27.5]	25.7 [24.2–27.9]	0.6	26.8 [24.5–28.1]	0.78
**6MWT**					
Pre-test Borg scale (median, IQR)	0 [0–1.75]	0 [0–0]	0.22	0.25 [0–1.25]	0.55
Post-test Borg scale (median, IQR)	3 [2–4.75]	2 [1–3]	0.15	4 [2–7]	0.36
Pre-test SpO_2_ (%, median, IQR)	97 [95.2–98]	97 [95–97]	0.23	95.5 [94–97]	**0.006**
Post-test SpO_2_ (%, median, IQR)	94 [91–95]	93 [95–92]	0.87	92 [89–95]	0.27
Pre-test HR (bpm, mean, SD)	75.4±19.1	77.5±12.9	0.67	81.5±13	0.21
Post-test HR (bpm, mean, SD)	93.3±17.8	93.8±18.1	0.94	96.4±20.4	0.66
6MWD (m, median, IQR)	445 [352.5–485]	460 [391.5–501]	0.37	310 [212–440]	**0.002**

**Table 5 jcm-14-08771-t005:** Multivariate backward logistic regression for GOLD categories classification. Model 1: comparison between GOLD A and GOLD_D_. Model 2: comparison between GOLD B and GOLD_D_. Model 3: overall model.

	AUC [95% CI]	OR [95% CI]	*p* Value
**Model 1**	0.81 [0.71–0.91]		**<0.0001**
Work exposure		0.03 [0.001–0.28]	**0.009**
Chest HRCT			
Consolidations		7.8 [1.3–74.7]	**0.04**
Centrilobular nodules		4.4 [1.2–18.2]	**0.03**
FVC		2.1 [0.96–4.9]	0.08
6MWD		0.99 [0.99–1]	0.1
**Model 2**	0.89 [0.82–0.95]		**<0.0001**
Work exposure		0.02 [0.0006–0.2]	**0.002**
BMI		0.9 [0.77–0.99]	0.05
FEV1		8.7 [2.7–37.2]	**0.001**
Chest HRCT			
Consolidations		13.7 [1.8–249.3]	**0.02**
**Model 3**	0.84 [0.76–0.91]		**<0.0001**
Work exposure		0.03 [0.001–0.2]	**0.005**
BMI		0.92 [0.004–7.7]	0.37
FVC		2.3 [1.2–4.6]	**0.01**
Chest HRCT			
Consolidations		5.9 [1.3–32.7]	**0.03**
Centrilobular nodules		3.5 [1.1–11.3]	**0.03**

## Data Availability

The original contributions presented in this study are included in the article/Appendix A. Further inquiries can be directed to the corresponding author.

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
