# Peer review of "Real-Life Characteristics of Patients with COPD and Discordant CAT-mMRC Questionnaires"

_jcm, 2025, doi:10.3390/jcm14248771_

Round 1
Reviewer 1 Report
Comments and Suggestions for Authors
The manuscript addresses an important challenge in COPD management by exploring patients with discordant COPD Assessment Test (CAT)- modified Medical Research Council(mMRC) scores, introducing the Global Initiative for Chronic Obstructive Lung Disease, discordant (GOLDD) group as a distinct and often misclassified phenotype. The study is well-designed and-conducted, using robust methods to highlight key differences in lung function, radiological findings, and occupational exposures for this subgroup. While recommendations include adding GOLDexacerbation (GOLD E) data and refining tables and figures, the research makes a significant contribution. It emphasizes the need for a more comprehensive, trait-based approach to COPD classification beyond symptom scores, offering practical clinical insights.
1. The authors conducted a robust, real-life, retrospective study that involved 222 COPD patients enrolled from Feb 2020 to Dec 2023. The methodology included spirometry, plethysmography, arterial blood gas (ABG)analysis, 6-minute walking 58 2. Test (6MWT), and high resolution computed tomography (HRCT) for comprehensive patient characterization. Adopting clear exclusion criteria ensured right patient cohort has been selected. Apppropriate use of statistical analyses method strengthens the quality and interpretation of data. 3. The results showed that GOLD D patients differ notably from GOLD A (CAT < 10 and a mMRC <1) and B (CAT ≥ 10 and a mMRC ≥ 2) groups. At baseline, 12.6% of patients fell into the GOLD D classification, characterized by lower occupational exposure to pollutants, intermediate lung function impairment (worse than GOLD A but better than GOLD B), and increased prevalence of HRCT-detected lung consolidations and centrilobular nodules. Logistic regression highlighted these features as independent predictors of GOLD Dcategorization. Moreover, GOLD classification showed instability over time, with 35.7% of GOLD D patients reclassified during follow-up. These findings suggest potential heightened susceptibility to infectious exacerbations that warrants further advanced study. 4. While the data seem informative, revisions to the comparison tables are needed for clarity and better interpretations. First, GOLD E data, representing 28.4% of the cohort, is missing, which limits comprehensive insights into comparisons with the most impaired group. Second, GOLD D data is repeated across adjacent columns in Tables 1-3, creating unnecessary redundancy. Additionally, a typographical error in Table 1 lists BMI with the unit Days, which should reflect kg/m². Finally, the IQR for Post-test SpO2 in Table 3 is incorrectly formatted and needs correction. 5. To improve clarity and impact, inclusionof GOLD E data in Tables 1-3, highlighting HRCT and 6MWT/ABG findings could be helpful. Making the tables more streamlined by consolidating GOLD D columns and adding separate P value columns are recommended. Correct errors in IQR formatting (Table 3). Enhance Figure 3 by labeling patient numbers and classification percentages for clarity. 6. This study has some notable limitations. The moderate sample size (N=222) and small follow-up group (N=95) limit the generalizability of results, especially for classification stability. Additionally, the lack of quantitative radiological scoring and absence of biomarker data restrict deeper insights into chest HRCT findings and possible inflammatory endotypes, highlighting important areas for future research.Minor
1. The manuscript is abbreviation-heavy and a table presenting all non-standard abbreviations and full explanations would be helpful. 2. Section 2.3: GOLD classification if presented in a tabular format will help the readers to reefer individual GOLD subgroups more conveniently.Author Response
Please see the attachment.

Reviewer 2 Report
Comments and Suggestions for Authors
In the manuscript, the authors present some real life characteristics of patients with COPD and they also analyzed some discordant CAT and mMRC questionnaires. The aim of the study was to To identify clinical, functional, and radiological characteristics associated with discordant CAT and MRC scores in patients with COPD. They included in the study 222 patients. It is a retrospective study. In my opinion, it is an interesting manuscript , that is well written. In order to improve the quality of the manuscript, some changes have to be done. My observations are :
- please include in the manuscript some date regarding the treatment that was used in this cases
- did you record any deaths in your study ? Please present this details.
Reviewer 3 Report
Comments and Suggestions for Authors
At the beginning, the importance of the topic should be emphasized. The authors presented all the necessary divisors and had only a 13% degree of text matching confirmed through iThenticate!
- The study contains Ethical approval, but it is important to indicate the date of approval along with the decision number!
- The statistical analysis was well conducted, as was the methodology.
All tables must be transferred to the results and the amount of text in the results reduced!
The discussion must be elaborated and the conclusion separated from the rest of the text!
Round 2
Reviewer 1 Report
Comments and Suggestions for Authors
The responses provided by the authors were difficult to match the original queries as the original comments have been truncated by the authors and the question numbers don’t exactly follow the original numerical order. This has created trouble to match the responses back to the original queries. Also the changes made in the manuscript following this reviewer’s comments have not been properly indicated that also created a hurdle to overcome while reviewing the revised version of the manuscript. The authors are advised to provide point-by-point responses to the questions by retaining the original question numbers.
The authors have provided satisfactory responses to all my questions/comments in the revised manuscript. The manuscript has now been much improved.
Reviewer 2 Report
Comments and Suggestions for Authors
In the manuscript, the authors tried to identify clinical, functional, and radiological characteristics associated with discordant CAT and MRC scores in patients with COPD. The manuscript has been reviewed before and the authors changed the manuscript according to the previous reviewers indications. Their comments are quite pertinent. Now, the quality of the manuscript has been improved. That is why, I think that this manuscript can be published in this form.
Reviewer 3 Report
Comments and Suggestions for Authors
No